# Federated Learning with Quantum Computing and Fully Homomorphic Encryption: A Novel Computing Paradigm Shift in Privacy-Preserving ML

**Siddhant Dutta**
DJ Sanghvi College of Engineering
siddhant.dutta180@svkmmumbai.onmicrosoft.com

**Pavana P Karanth**
GSSS Institute of Engineering
pavana.karanth17@gmail.com

**Pedro Maciel Xavier**
Davidson School of Chemical Engineering
Purdue University
pmacielx@purdue.edu

**Iago Leal de Freitas**
Davidson School of Chemical Engineering
Purdue University
ilealdef@purdue.edu

**Nouhaila Innan**
eBRAIN Lab, NYU Abu Dhabi (NYUAD)
Center for Quantum and Topological Systems
NYUAD Research Institute
nouhaila.innan@nyu.edu

**Sadok Ben Yahia**
The Maersk Mc-Kinney Moller Institute
University of Southern Denmark
say@mmmi.sdu.dk

**Muhammad Shafique**
eBRAIN Lab, NYU Abu Dhabi (NYUAD)
Center for Quantum and Topological Systems
NYUAD Research Institute
muhammad.shafique@nyu.edu

**David E. Bernal Neira**
Davidson School of Chemical Engineering
Purdue University
dbernaln@purdue.edu

## Abstract

The widespread deployment of products powered by machine learning models is raising concerns around data privacy and information security worldwide. To address this issue, Federated Learning was first proposed as a privacy-preserving alternative to conventional methods that allow multiple learning clients to share model knowledge without disclosing private data. A complementary approach known as Fully Homomorphic Encryption (FHE) is a quantum-safe cryptographic system that enables operations to be performed on encrypted weights. However, implementing such mechanisms in practice often has significant computational overhead and can expose potential security threats. Novel computing paradigms, such as analog, quantum, and specialized digital hardware, present opportunities for implementing privacy-preserving machine learning systems while enhancing security and mitigating performance loss. This work instantiates these ideas by applying the FHE scheme to a Federated Learning Neural Network architecture that integrates both classical and quantum layers.

## 1   Introduction

With the widespread deployment of Machine Learning (ML) applications, the level of direct human-machine interaction has increased rapidly. This surge has raised concerns and increased user awareness about the capabilities and limitations of these technologies. As the impact of Machine Learning (ML)–based gadgets becomes more relevant in the public discourse, governmental agencies begin

38th Second Workshop on Machine Learning with New Compute Paradigms at NeurIPS 2024(MLNCP 2024).

to develop and implement regulatory policies regarding the fair use and overall protection of user data. For example, the 2016 EU Data Regulation Act [1] and the Brazilian LGPD from 2018 [2] establish guidelines for the processing of personal data, setting the security requirements for safe information storage and delimiting the scope of use of these data. To address this issue, the Federated Learning (FL) framework was proposed [3, 4] as a mechanism for coordinating multiple independent clients that cooperate in a shared learning task by transmitting only the "knowledge" of the trained model to their peers while keeping private data stored locally. In contrast to conventional ML, where data is often centralized in a single server for model training, this distributed approach is suitable for addressing data privacy concerns. Each independent client shares their model updates with a server responsible for combining and broadcasting the aggregated model back to its clients. This allows one to benefit from the insights produced through other clients' data without ever having direct access to it. However, these benefits are not free. The siloing of the data compromises the speed with which "knowledge" diffuses through the network, affecting the efficiency of training in the form of computational overheads and communication inefficiencies [5]. Moreover, exposing model data to potentially vulnerable communication channels between clients and the server could defeat FL's original privacy goal. Even with the distribution of data for each client in an FL framework, the privacy of this data can be threatened by the interception of messages between the clients and the server. Once these messages are intercepted, the original private data can be inferred. To this end, an extra layer of quantum-safe privacy protection is implemented by encrypting the model updates before reaching the central server so that its aggregation operations are performed on this data without decrypting it. This technique is known as Fully Homomorphic Encryption (FHE) [6]. Implementing this technique prevents the server from accessing direct model updates, which prevents any potential intercept of the messages sent by the clients. Clients can then trust that the aggregation technique is performed without exposing their local data or the resulting learning model. Since the encryption occurs before the model updates are communicated, each client's local ML model is not restricted, yet it is protected by FHE. This layer of protection comes at the expense of higher resource consumption to manipulate encrypted model updates [6].

New techniques must be implemented to address all the considerations of data privacy on the scale on which these FL models will be deployed. Tackling this challenge requires improvements in how efficiently each client can learn individually, reducing the number of communication rounds, and in how to encrypt the messages exchanged with a server. We consider that novel computational paradigms can be the answer to these challenges. In particular, we claim that each client can address their learning tasks with enhanced architectures that leverage this hardware. The compositional nature of deep learning models allows some of its layers to be implemented using new computing paradigms such as photonic [7], neuromorphic [8, 9] or quantum computers [10]. These machines consist of specialized hardware with promising computational speedup capabilities relevant to ML, such as matrix multiplication or calculating gradients.

One paradigm that has received particular attention in the context of privacy is *Quantum Computing* [11]. Quantum computing (QC) is the processing of information using phenomena explained through quantum mechanics. The basic information unit for QC is the quantum bit or *qubit*, which can be a superposition of the states 0 or 1. Processing over qubits subject to quantum mechanics allows one to accelerate specific computational tasks, even exponentially [10]. Quantum computers, that is, devices capable of implementing quantum computations, are still limited in size and capabilities primarily concerned with handling unintended interactions with their environment [10], which has the same effect of projecting the quantum states onto a classical one, known as *measurement*. However, they are steadily gaining traction, with the potential expectation that in the future they can surpass classical machines [12, 13] in tasks related to combinatorial optimization [14] and cryptography [15]. This opens the doors for hybrid ML algorithms that, acknowledging and accounting for the limitations of each computational paradigm, take advantage of the classical and quantum layers [16, 17].

## 2  Related Work

Since its inception, FL has focused on communication-efficient learning with applications to data privacy [3]. Subsequent research has expanded on this foundation, with different applications [4, 18, 19] and addressing challenges arising from practice such as training over non-IID data [20]. FL combined with FHE has gained prominence as a privacy-preserving approach to ML [21]. In particular, there has been an increase in applications in healthcare [22, 23], where preserving privacy

is crucial when working with data from medical diagnosis and imaging [24–26]. Recent work has focused on tackling the computational inefficiencies inherent in FHE [27, 28].

Another prominent area consists of adding quantum computing layers to the distributed clients' architecture, giving rise to an extension of Quantum Machine Learning (QML) [29, 30] known as Quantum Federated Learning (QFL) [16, 31]. Special attention has been paid to QFL on quantum [32] or decentralized data [33]. Extensions of QFL considering other classical ML architectures, such as convolutional networks [34, 35], have been proposed. Applications of QFL in healthcare have been explored using existing quantum hardware [36] or classical simulations of quantum computers [23]. Other applications in finance [37] and Internet-of-Things (IoT) security [17] have been explored. There are also previous developments for integrating it with encrypted weights [38]. Finally, as a novel technology, it faces implementation and resource allocation challenges [39, 40].

## 3  Problem Description and Methodology

Despite its applications in distributed ML, classical FL still faces challenges. Among these are communication bottlenecks across large networks [41], privacy concerns during model updates [42], especially sensitive data such as those found in healthcare applications [23], and computational inefficiencies due to training large datasets on devices with limited resources [43]. QFL is a distributed learning paradigm capable of tackling some of these challenges. It consists of clients capable of accessing quantum computers that collaboratively train a global model while communicating with a centralized classical server. Since the local computations found in FL tend to be smaller than centralized ML datasets, it becomes a great use case for the still resource-constrained quantum devices available today [10, 44]. Furthermore, QFL is capable of using quantum-enhanced communication protocols that offer inherent privacy advantages over the classical ones [15].

In a standard neural network, the weights are iteratively updated by gradient descent. The weight update at time step $t+1$ is $W_i^{t+1} = W_i^t - \eta \nabla \mathcal{L}(W_i^t)$, where $W_i^t$ denotes the weight at time $t$ for client $i$, $\eta$ is the learning rate, and $\nabla \mathcal{L}(W_i^t)$ is the loss function gradient with respect to the weight. This update minimizes the loss by adjusting the weights accordingly. However, in a scenario using FHE, the process changes since weights are encrypted on each client. FHE allows performing the same calculations on the encrypted weights $\mathcal{E}(W_i^t)$, generating new encrypted weights as $\mathcal{E}(W_i^{t+1}) = \mathcal{E}(W_i^t) - \eta \nabla \mathcal{L}(\mathcal{E}(W_i^t))$, leveraging FHE's ability to perform operations directly on encrypted parameters. FL with FHE aggregates encrypted weights across multiple clients without revealing

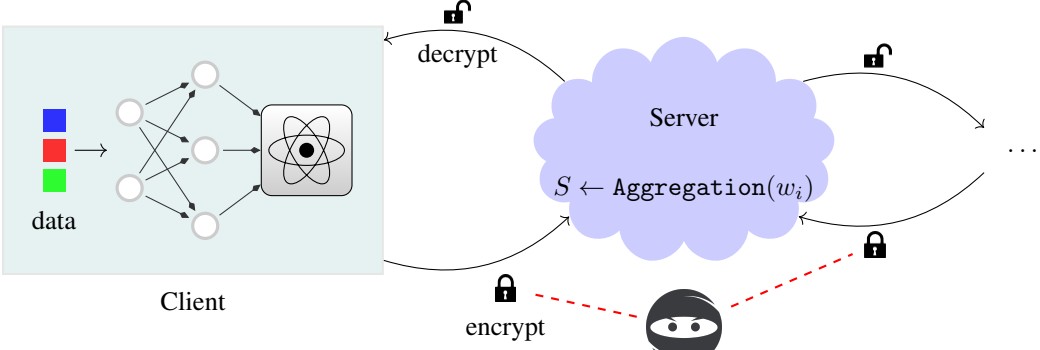

Figure 1: The client nodes utilize local data to train a model that incorporates both classical and quantum layers. After training, these models are encrypted and transmitted to a central server for aggregation into the average of all encrypted models. This new global model is then distributed to all clients. This global model is decrypted on each client, initiating another round of training. Intruders to the client-server communication would only intercept of quantum-safe encrypted models.

individual weights. The server operates on encrypted weights as $\mathcal{E}(S^{t+1}) = \sum_{i=1}^{N} c_i \mathcal{E}(W_i^{t+1})$, where $c_i$ is a proportion parameter for the $i$-th client. Upon receiving the updated weights, the clients decrypt them and set their weights for the next model as $W_i^{t+1} = \mathcal{E}^{-1}(\mathcal{E}(S^{t+1}))$.

QFL with FHE works by training and encrypting the models for all clients in parallel. At the same time, a centralized server receives models, aggregates them, and redistributes the new model to all clients according to the procedure just described. This process repeats until convergence or meeting

any other stopping criterion. Algorithm 1 gives a pseudocode description of the procedure, and Fig. 1 shows a high-level visualization.

---

**Algorithm 1** Quantum Federated Learning with Fully Homomorphic Encryption

---

 1: **Require:**
 2:     $ctx$: Fully homomorphic encryption context
 3:     $N$: Number of federated clients
 4:     $params$: Encryption parameters
 5:     $G$: Quantum gate set
 6:     $D$: Parameterized Quantum Circuit (PQC) depth
**Ensure:** Aggregated global model $\mathbf{w}_g$
 7: **Initialization:**
 8: Generate CKKS context $ctx \leftarrow$ CKKSContext($params$)
 9: Generate Galois keys for rotations keys $\leftarrow ctx$.generate_galois_keys()
10: Initialize global QNN model $\mathbf{w}_g \leftarrow$ InitializeQNN($D, G$)
11: **Client-Side QNN Training and Encryption:**
12: **for** each client $k \in \{1, \ldots, N\}$ **in parallel do**
13:     Prepare quantum dataset $\mathcal{D}_k \leftarrow$ PrepareQuantumDataset($k$)
14:     Train local QNN $\mathbf{w}_k \leftarrow$ TrainQNN($\mathcal{D}_k, \mathbf{w}_g, D, G$)
15:     Quantize and encrypt the local model $\mathbf{w}_k^{enc} \leftarrow$ Encrypt(Quantize($\mathbf{w}_k$), $ctx$)
16:     Send encrypted model $\mathbf{w}_k^{enc}$ to the server
17: **end for**
18: **Server-Side Aggregation:**
19: Initialize $S \leftarrow 0$                                            ▷ Accumulator for weighted sum
20: $n_{\text{total}} \leftarrow \sum_{k=1}^N n_k$                        ▷ Total number of samples across all clients
21: **for** each client $k \in \{1, \ldots, N\}$ **do**
22:     Receive $\mathbf{w}_k^{enc}$ from client $k$
23:     Aggregate encrypted weights $S \leftarrow S + \mathbf{w}_k^{enc} \cdot \frac{n_k}{n_{\text{total}}}$
24: **end for**
25: **Client-Side Decryption and Global Model Update:**
26: **for** each client $k \in \{1, \ldots, N\}$ **in parallel do**
27:     Decrypt aggregated model $\mathbf{w}_g \leftarrow$ Decrypt($S$, secret_key)
28:     Update global QNN model $\mathbf{w}_g$ on the client
29: **end for**
30: **PQC Update:**
31: Adjust PQC parameters and architecture $\mathbf{w}_g \leftarrow$ OptimizePQC($\mathbf{w}_g, D, G$)
32: **Model Distribution:**
33: **for** each client $k \in \{1, \ldots, N\}$ **in parallel do**
34:     Send global model $\mathbf{w}_g$ to client $k$
35: **end for**
36: **Repeat** from step 11 until maximum communication rounds
37: **return** $\mathbf{w}_g$

---

## 3.1 Quantum Neural Network Initialization and Client-Side Training

The Quantum Neural Network (QNN) is initialized by constructing a variational Parameterized Quantum Circuit (PQC) with a specified depth $D$ and a set of quantum gates $G$. An example of this PQC that we used in our illustrative test cases is given in Fig. 2. This variational PQC, which will be trained on quantum data, uses parameters such as quantum gate angles and biases that can either be initialized randomly or based on pre-trained values. After initialization, each client prepares its quantum dataset $\mathcal{D}_k$, which could be based on local quantum measurements or preexisting datasets. Using this data, the client trains its local QNN through variational quantum algorithms. Once training is complete, the model weights are quantized to a fixed precision format to reduce the size of the encrypted parameters before encrypting the weights for secure transmission to the central server. The CKKS scheme is configured with a polynomial modulus degree of 8192, which defines the ring size $\mathbb{Z}[X]/(X^n + 1)$ with $n = 8192$, providing a security level of 128 bits. The coefficient modulus is split into four primes with bit sizes [60, 40, 40, 60], resulting in a total modulus size of 200 bits, balancing security and computational efficiency. The global scaling factor of $2^{40}$ ensures sufficient precision for fixed-point arithmetic.

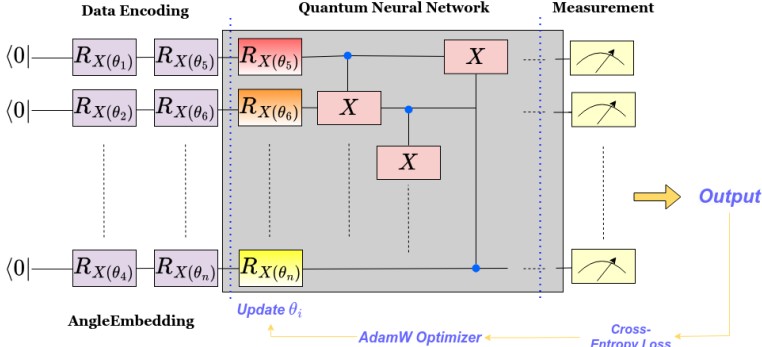

Figure 2: Depiction of quantum circuit used at the client's level for QFL. Input data is encoded into a quantum state using angle embedding via parameterized rotation gates $R_X(\theta_i)$. The encoded quantum states are then processed by a PQC, where the weights & parameters of the PQC are encrypted using FHE. The encrypted quantum states undergo operations involving parameterized rotation gates & Controlled-NOT gates, facilitating entanglement & complex quantum state manipulation in the encrypted domain. After measurement, classical outputs are obtained, and the Cross-Entropy loss is computed. The encrypted parameters $\theta_i$ are updated during training using a classical optimizer.

### 3.2 Server-Side Aggregation and Global Model Update

On the server, encrypted QNN weights from all clients are aggregated using a weighted summation method, where each client's contribution is proportional to the size of their data set. This ensures that the global model reflects each client's training effort. Further optimizations to the PQC, such as modifying the depth $D$ or adjusting the gate set $G$, may be performed to enhance model performance. After these updates, the global model is distributed back to the clients, and this iterative process continues until convergence, ultimately resulting in a privacy-preserving QNN model.

## 4    Computational Results

Training times for FHE-FedQNN models are notably extended due to the combined computational demands of quantum simulation and FHE. This increased duration is particularly evident with datasets like CIFAR-10, where the use of 6 qubits to represent 10 classes adds to the computational burden. Similarly, Brain MRI, which requires 4 qubits for 4 classes, and PCOS, with only 2 qubits for 2 classes, reflect varying computation times based on the number of qubits utilized. The choice of batch size is adapted to the dataset's size. For CIFAR-10, with a substantial number of images (48k training and 12k testing), a batch size of 128 is used. In contrast, smaller datasets such as Brain MRI and PCOS, with 5.7k/1.3k and 2.56k/0.64k samples, respectively, used smaller batch sizes of 32. These adjustments help to optimize computation within the FL framework according to the size of the dataset.

Concerning Table 1, although the introduction of FHE results in computational overhead, the impact on test accuracy for FHE-FedQNN models is minimal. The difference compared to standard FedQNN models is around 1-2%, suggesting that the benefits of enhanced data security and quantum processing can outweigh this slight accuracy trade-off. Upon evaluating the FHE-FedQNN model, it was observed that there was improved performance in the PCOS dataset, resulting in a 4% gain in classification accuracy. This progress suggests that the FHE scheme could potentially assist the model in managing the noise introduced by encryption, thereby improving its generalization capabilities. Despite the increased test loss in the FHE-FedQNN model, which is likely due to noise amplification from FHE, the quantum model demonstrates superior generalizability. All models achieved near-optimal training accuracies, a typical outcome in FL settings since each client trains on a subset of data. However, test accuracy, which is measured on a separate set of test images, more accurately reflects the performance of the aggregated federated model.

## 5    Discussion

In the end, the more complicated architecture of FL with FHE induces a trade-off from speed to privacy. We have shown that new computing paradigms, and in particular a quantum computer, can be

Table 1: Comprehensive performance analysis of Fully Homomorphic Encryption-Enabled Federated Quantum Neural Networks (FHE-FedQNN) and Federated Neural Networks (FHE-FedNN) versus their Standard counterparts (non-FHE) on CIFAR-10, Brain MRI, and PCOS Datasets. Each model was trained over 20 Rounds with 20 clients and 10 epochs per round.

| Dataset | FHE-FedQNN Models | | | | Standard FedQNN Models | | | |
|---|---|---|---|---|---|---|---|---|
| | Train. Acc. | Test. Acc. | Test. Loss | Time (min) | Train. Acc. | Test. Acc. | Test. Loss | Time (min) |
| **CIFAR-10** [45] | 99.10% | 70.12% | 1.240 | 156.5 | 97.15% | 72.16% | 1.202 | 151.5 |
| **Brain MRI** [46] | 99.60% | 88.75% | 0.360 | 116.5 | 100.00% | 89.71% | 0.338 | 110.6 |
| **PCOS** [47] | 100% | 70.15% | 1.09 | 87.2 | 100% | 66.19% | 0.611 | 70.9 |

| Dataset | FHE-FedNN Models | | | | Standard FedNN Models | | | |
|---|---|---|---|---|---|---|---|---|
| | Train. Acc. | Test. Acc. | Test. Loss | Time (min) | Train. Acc. | Test. Acc. | Test. Loss | Time (min) |
| **CIFAR-10** [45] | 100% | 68.53% | 1.322 | 136.4 | 100% | 71.09% | 1.257 | 128.9 |
| **Brain MRI** [46] | 100% | 88.4% | 0.402 | 98.4 | 100.00% | 90.36% | 0.298 | 89.3 |
| **PCOS** [47] | 100% | 64.11% | 1.379 | 84.3 | 100% | 65.37% | 0.813 | 68.6 |

used in these ML models as a tool to accelerate local computations. The small-scale clients' models in FL are more amenable to the limits encountered on current quantum devices. However, considering the current quantum hardware scale, this approach still has limitations. These can be mitigated in some cases by classical simulation of quantum systems via tensor networks [23, 48], although only practical until a certain scale. In the future, advances in quantum hardware, qubit error correction, and encryption techniques are expected to make QFL practical for real-world applications.

For future work, an in-detailed study of the loss flow & the gradients flow rate is necessary to provide conclusive evidence on the performance impact of QNNs integrated with FHE. This investigation will help quantify the trade-offs between encryption and model accuracy. Additionally, exploring more advanced quantum circuit designs is crucial to mitigate the issue of barren plateaus, which can hinder optimization and training in quantum neural networks. These efforts will enhance both the efficiency and scalability of FHE-enabled QNNs. All code for the results in the methodology is open source and available in the repository `https://github.com/elucidator8918/QFL-MLNCP-NeurIPS`.

## 6 Perspectives

FL has emerged as a viable technology for machine learning in domains where data privacy is important. Challenges related to training efficiency and vulnerability to eavesdropping have spurred a number of developments, including FHE. Leveraging the composability of current deep learning methods, some proposals have integrated classical and novel computational paradigms to satisfy the ever-growing requirements of FL applications. In particular, quantum computing has been successfully integrated with FL in this work and others [23, 31, 34, 40]. Our contribution was to show the potential of combining FHE with quantum FL and provide an implementation of these methods that is replicable on classical computers through efficient simulation of quantum circuits. The results obtained suggest that incorporating both quantum layers and FHE does not significantly increase the training time, and in some cases, it even improves the learning metrics. More importantly, it shows how new computing paradigms can already aid in relevant ML tasks.

These novel computational paradigms still have significant untapped potential. We highlight that FL can be the meeting point of two branches of quantum information sciences: quantum computing and quantum communication. To achieve exponential speedups using QML, it has been shown that one can operate directly over quantum data, without the need for encoding [33, 49]. At the same time, the advantages of quantum communication arise only when transmitting qubits. Exploring the simultaneous usage of both technologies presents a fascinating application of this technology, with federated and machine learning being the use case that requires them together.

## Acknowledgment

This work was supported in part by the NYUAD Center for Quantum and Topological Systems (CQTS), funded by Tamkeen under the NYUAD Research Institute grant CG008, and the Center for Cyber Security (CCS), funded by Tamkeen under the NYUAD Research Institute Award G1104. P.M.X., I.L.d.F, and D.E.B.N graciously acknowledge the support of the Purdue Davidson School of Chemical Engineering startup grant.

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
