# OpenReview forum: "Federated Learning with Quantum Computing and Fully Homomorphic Encryption: A Novel Computing Paradigm Shift in Privacy-Preserving ML"
_NeurIPS.cc/2024/Workshop/MLNCP — MLNCP Poster_

### Official Review · Reviewer_oS8G · 2024-10-03
**Review for Federated Learning with Quantum Computing and Fully Homomorphic Encryption: A Novel Computing Paradigm Shift in Privacy-Preserving ML**

**Rating:** 4
**Confidence:** 2

**Review:**

This paper presents a methodology that combines fully homomorphic encryption and quantum federated learning. Both of these techniques can improve privacy preservation for federated learning. Fully homomorphic encryption enables clients to securely inform server updates to encrypted model weights. These updated encrypted weights at the server can be sent back to the clients to update the client models. This process of operating directly on encrypted weights enables clients to cooperatively train models with a centralized server without exposing any local data. Quantum federated learning enhances typical federated learning approaches with the benefits of quantum computing. This can provide improved computational efficiency for some types of problems and additional privacy improvements.

This paper does show that federated networks can effectively be trained with and without quantum neural networks and with and without FHE, but it is not clear to me how the benefits of quantum neural networks and FHE work together. The paper seems to suggest additional privacy benefits may be produced by this combination, but does not provide sufficient analysis of this. As training networks with this combination is the key contribution of this paper, it seems necessary that the benefits of this combination is assessed/discussed in detail.

---

### Decision · Program_Chairs · 2024-10-10

Accept (Poster)